# Simulation Analysis of the Dispersion of Typical Marine Pollutants by Fusion of Multiple Processes

Xueqing Guo [1], Yi Liu [1], Jian-Min Zhang [2,3], Shengli Chen [1], Sunwei Li [1] and Zhen-Zhong Hu [1,2,*]

1  Institute for Ocean Engineering, Tsinghua Shenzhen International Graduate School, Shenzhen 518055, China; gxq20@mails.tsinghua.edu.cn (X.G.); yiliu20@mails.tsinghua.edu.cn (Y.L.); shenglichen@sz.tsinghua.edu.cn (S.C.); li.sunwei@sz.tsinghua.edu.cn (S.L.)
2  Institute for Ocean Engineering, Tsinghua University, Beijing 100084, China; zhangjm@tsinghua.edu.cn
3  Department of Hydraulic Engineering, Tsinghua University, Beijing 100084, China
*  Correspondence: huzhenzhong@tsinghua.edu.cn

**Abstract:** The rapid development of coastal economies has aggravated the problem of pollution in the coastal water bodies of various countries. Numerous incidents of massive-scale marine life deaths have been reported because of the excessive discharge of industrial and agricultural wastewater. To investigate the diffusion of typical pollutants after discharge, in this study, a multi-process fusion simulation analysis model of pollutants under the action of ocean currents was established based on the concentration analysis method. Furthermore, key technologies involved, such as the parameter value, data selection, and visualization, were investigated. The iterative analysis and programming realization of three independent sub-processes, such as pollutant diffusion and transport, and the drift path and concentration distribution of pollutants after their discharge into the sea, were visualized. The case study revealed that the increase in the concentration of pollutants in the ocean was affected by the diffusion sub-process, and the transport sub-process plays a critical role in the long-distance transport of pollutants. The proposed method can provide technical support for marine environmental risk assessment and dynamic tracking of marine pollutants.

**Keywords:** typical pollutants; concentration analysis; multi-process; numerical simulation

## 1. Introduction

With the rapid development of the global economy, the impact of human activities on the surrounding environment has become stronger, which has intensified the problems of environmental pollution and ecological destruction. Because the ocean contains abundant mineral, chemical, biological, and energy resources, sustainable development is critical in the economic, military, scientific, and environmental spheres. However, with continuous development in the marine field, numerous human-induced activities, such as the discharge of industrial and agricultural wastewater and the excessive emission of domestic sewage, have adversely affected marine ecological environment. Pollutants in the ocean have a cumulative negative effect on marine organisms and the marine ecological environment. Therefore, swiftly identifying pollutant diffusion paths and sources at the initial stage of pollution dispersion is a critical research topic both domestically and internationally. The marine pollutant dispersion model provides excellent theoretical and technical support for the rapid emergency response to marine pollution and the traceability of pollutants.

Human activities are negatively affecting the ocean. In particular, the excessive discharge of industrial and agricultural wastewater has severely damaged the marine ecological environment. Industrial and agricultural wastewater typically contain large amounts of heavy metal pollutants, such as mercury, cadmium, lead, and copper [1]. Most of these pollutants have low solubility and are easily absorbed by marine plants and animals, which greatly affect the survival of organisms in nearshore waters and the development of the fishing industry. For example, mercury can accumulate in seawater,

sediment, and fish, and cause poisoning of the food chain, as observed in the 1956 Minamata disease incident in Japan [2]. Cadmium cannot be ingested by the human body, and long-term consumption of cadmium-contaminated fish or algae can lead to "Itai-itai disease," which causes severe bone deformities, intense pain, height shrinkage, and brittle bones [3]. Radioactive pollutants discharged into the ocean are highly dangerous. They are generally easily absorbed by sediments, accumulate on the seafloor, and can enter the human body through the food chain, eventually leading to genetic damage or mutations in human DNA. Severe marine pollution irreversibly affects a country's ecological environment and economic development. Therefore, mitigation efforts should be focused on the prevention, monitoring, and emergency response to enhance marine environmental management [4]. In addition, proactive measures and solutions should be employed to protect the marine ecological environment.

Based on the diffusion characteristics, pollutant diffusion simulations are typically categorized into two types, namely point source pollutant diffusion and nonpoint source pollutant diffusion. Nonpoint source pollutant diffusion simulation methods are typically used for plain river networks with complex tributary distribution and pollutant sources. Currently, some studies [5–7] conducted in-depth investigations of the numerical simulation of river and groundwater systems and pollutant source tracing technology. Makri et al. [8] conducted an in-depth study on how the hydrological characteristics of groundwater systems affect the diffusion and attenuation of heavy metal pollutants such as benzene. However, microbial pollutants in the ocean originate from the fixed discharge of industrial and agricultural wastewater; therefore, point source pollutant diffusion simulation methods are widely adopted. Scientists summarized suitable diffusion simulation methods based on various diffusion characteristics. The present study introduces the commonly used numerical simulation methods for point source pollutant diffusion, without delving into machine learning methods that are commonly employed in nonpoint source pollutant diffusion simulations.

In 1971, on the oceanic scale, Stewart et al. [9] developed a model for predicting the concentration of coliform bacteria in downstream marine areas based on measurements of nearshore water flow and estimates of sewage turbulent diffusion. The model was used to assess future ocean discharge outlet settings. In 1978, Lam et al. [10] discussed the application of statistical and deterministic models in simulating the diffusion of coastal discharge outlets, which were used to assess the continuous discharge of wastewater into the sea at specific locations. Additionally, another study [11] combined the Lagrangian particle-tracking method with the Eulerian convection-diffusion model of pollutants and focused on the Mediterranean Sea to simulate the motion of plastic particles over two years and the exchange of pollutants with the surrounding environment. These studies mostly used two-dimensional tidal current motion continuity equations, momentum equations, and pollutant transport equations, which involve numerous formulas and require multiple boundary and initial parameters. Moreover, these methods are mainly suitable for small-scale diffusion simulations and have limitations for large-scale spatial diffusion simulations.

The improvement of observation and calculation conditions has resulted in the emergence of refined diffusion models gradually in recent years. Some researchers [12] developed a Fickian particle-tracking diffusion coupling model based on fractional Brownian motion and geographic information systems to predict pollutant drift. Gruber et al. [13] used eddy-resolving models to simulate the development of ocean acidification in various emission scenarios in California Current waters, providing support for policy formulation. Other studies [14,15] used the fully coupled global climate model CESM to ensure that the oceanic eddy processes in the prediction model are close to the current observation conditions. Ryan et al. [16] used a three-dimensional hydrodynamic model and observation data to validate a far-field pollutant diffusion model in Geographe Bay, Western Australia, and determined the effect of treated wastewater on the marine environment under various emission scenarios. Souza et al. [17] used the Delft3D far-field model to analyze and evaluate the sewage diffusion situation of a marine discharge outlet in Santos

City under various sewage treatment conditions and discharge outlet lengths, providing highly intuitive visualization analysis results. Using the three-dimensional ocean model FVCOM, Yuesong et al. [18] simulated the three-dimensional diffusion process of pollutants in the Yangtze River Estuary waters and studied its impact range and conducted ecological assessments.

To simulate the spread of radioactive materials in the marine environment, Johannessen et al. [19] proposed a set of numerical modeling techniques. Mesoscale mixing in the ocean strongly affects the simulation of radioactive pollutant transport. Another study [20] proposed a novel surface diffusion coefficient dataset based on long-term global velocity observations to accurately describe the mesoscale mixing rate in the oceans. In 2011, Prants et al. [21] used bivariate cubic spatial interpolation and Lagrangian time interpolation to numerically simulate the diffusion of Fukushima's radioactive nuclides in the ocean. This study was the first to conduct an oceanic-scale diffusion simulation of nuclear leakage accidents. In 2012, the German Ocean Agency [22] used a global ocean circulation model to estimate the long-term dispersion process of Cs-137 under mesoscale eddies and used it to simulate Cs-137 concentration distribution in the Pacific Ocean within 10 years; the established model is a widely recognized dynamic model. Using a high-resolution global coastal nested ocean model, Lai et al. [23], in 2013, reproduced the Fukushima earthquake tsunami, coastal flooding, and initial diffusion of the radioactive nuclide Cs-137. In 2022, the diffusion of radionuclides in the ocean within ten years using the radionuclide diffusion model was proposed for the Fukushima nuclear wastewater discharge plan, which attracted considerable attention both domestically and internationally.

Many difficulties, such as low-resolution ocean currents and other environmental data [24] as well as the difficulty of accurately representing internal flow fields through control equations or numerical methods, are encountered in simulating pollutant diffusion in the ocean [25–27]. Although studies proposed deterministic simulation methods for wave motion [28,29], these studies mostly targeted specific waveforms and are not suitable for large-scale spatiotemporal pollutant diffusion simulation. Furthermore, research on visualization technology for specific marine environments such as flow fields is still in the preliminary stage, and the selection of grid division and data preprocessing methods requires further exploration. Extensive research on multi-dimensional visualization technology for marine environments is needed [30–35].

To effectively simulate the diffusion process and concentration distribution of typical marine pollutants, such as heavy metals and radioactive materials, after their discharge into the ocean, in this study, a multiscale spatio-temporal pollutant diffusion analysis method is proposed. The proposed method can help in analyzing the diffusion mechanism of pollutants and is suitable for various sea areas worldwide. Through visual programming simulation, an open-source ocean current data set was used for the dynamic predictions of pollutant diffusion processes and distribution patterns. This study offers significant implications for the early prevention and control of pollutants and the formulation of emission policies.

In this study, first, the applicability and accuracy of the HYCOM ocean current dataset involved in the diffusion model were validated in the simulated sea area by comparing them with the reanalysis data of the ocean current field [36] obtained from the Hong Kong and Macau Ocean Research Center. Then, four randomly selected discharge outlets were used to simulate and analyze pollutant dispersion in the relevant marine areas to provide modeling and technical support for pollutant prevention and control, traceability, and other related tasks.

## 2. Multi-Process Fusion Simulation Analysis Method of Marine Pollutants

In static water, the diffusion velocity of pollutants is consistent in all directions; however, local water pollutants in rivers and oceans exhibit diverse movements. This study decomposed the basic diffusion process of pollutants into two approximately independent sub-processes, namely diffusion and transport. Diffusion refers to the transfer of pollutants

from high-concentration areas to low-concentration areas under concentration gradients, and this process is widely influenced by molecular diffusion, eddy diffusion under local turbulence, and dispersion caused by uneven cross-sectional flow velocity. Transport refers to the directional movement of pollutants with ocean currents, which changes the position of the pollutants without altering their concentration. Considering this, adding a corresponding radioactive pollutant decay coefficient term can satisfy the requirements of a typical pollutant diffusion simulation. During different steps of the analysis, the concentration distribution changes caused by each sub-process were considered separately, which were then superimposed to obtain a complete diffusion process, as displayed in Figure 1.

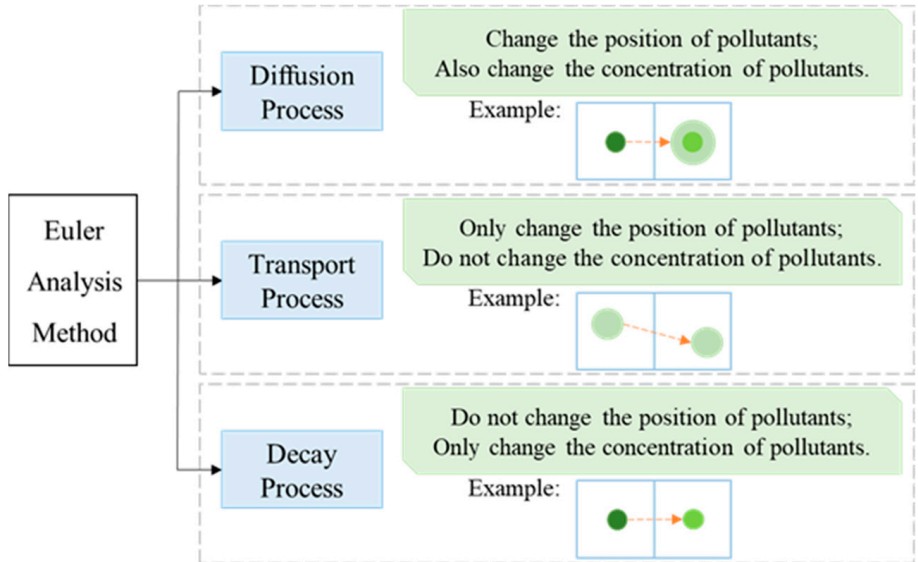

**Figure 1.** Main research content. (The depth of the circle color indicates the level of concentration.)

The finite element concept was adopted to discretize the simulation area into a series of small square grids. The concentration changes of the pollutants in each grid were calculated at each time step to obtain the pollutant concentration distribution at a specific time. The methods for each sub-process are as follows.

### 2.1. Pollutant Diffusion Process

The process of diffusion is characterized by the change in the pollutant concentration being proportional to the concentration gradient, and their ratio is called the diffusion coefficient, which is expressed in the form of Fick's law [37] as follows:

$$J = -D\frac{\partial c}{\partial x} \tag{1}$$

In Equation (1), $D$ is the diffusion coefficient, $\partial c/\partial x$ is the concentration gradient, and $J$ is the diffusion flux, i.e., the total amount of pollutants passing through a unit area in a unit time.

Considering a two-dimensional diffusion problem in which pollutants are uniformly distributed along the height, we first divided the two-dimensional plane into uniform square grids, with the grid side length of $\Delta x = \Delta y$. If the pollutant concentrations at the centers of two adjacent small squares are assumed to be $c_1$ and $c_2 (c_1 \geq c_2)$, the difference is denoted by $c_d = c_1 - c_2$. Assuming a constant concentration gradient between the center points of the two squares, we obtained the following expression:

$$\frac{\partial c}{\partial x} = \frac{c_d}{\Delta x} \tag{2}$$

It can be assumed that the concentration values of the two grids do not change significantly, and the concentration gradient remains constant within the time step $\Delta t$. Then, the total amount of pollutants passing through the contact surface of the two grids can be determined using the following equation:

$$\Delta n = |J \times S \times \Delta t| = D \frac{c_d}{\Delta x} \times S \times \Delta t \tag{3}$$

where $S$ is the area of the contact surface between the two grids. Assuming that pollutants are uniformly distributed along the height of the contact surface $h$, we obtain the following expression: $S = h \times \Delta y$.

Because two adjacent grids have the same volume and mass change, i.e., the increase in the mass of pollutants in cell 2 is equal to the decrease in the mass of pollutants in cell 1, the concentration changes in the two grids also remain the same, which can be expressed as follows:

$$\Delta c = \frac{\Delta n}{V} = \frac{D \frac{c_d}{\Delta x} \times h \times \Delta y \times \Delta t}{\Delta x \times \Delta y \times h} = \frac{D c_d \Delta t}{(\Delta x)^2} \tag{4}$$

That is:

$$\frac{\Delta c}{c_d} = D \frac{\Delta t}{(\Delta x)^2} \tag{5}$$

According to the equation, when the diffusion coefficient $D$ remains unchanged, after giving the grid side length $\Delta x$ along a certain direction and the simulation time step $\Delta t$, $\Delta c / c_d$ will be a constant value, which can be expressed as follows:

$$k = \frac{\Delta c}{c_d} = D \frac{\Delta t}{(\Delta x)^2} \tag{6}$$

$k$ value represents the simulation speed of the diffusion process. The larger the $k$ value, the fewer the simulation steps required to reach a uniform concentration state. Because this study assumed that the diffusion of substances is related only to the contact surface, the length of the contact surface in the diagonal direction of the grid was 0, i.e., no direct material transfer occurred between the four diagonal grids and the grid. Therefore, in the two-dimensional diffusion problem, it can be assumed that four contacting grids were placed around each grid. Similarly, when the three-dimensional space was expanded, the number of contacting grids around each grid increased to six. Therefore, to ensure the convergence of the simulation results, i.e., to avoid negative concentration values, in the two-dimensional diffusion problem, the equation should satisfy the condition $k \leq 1/(4+1) = 0.2$. Collectively, diffusion is related to the relative concentrations and not the absolute concentrations.

To determine the effect of various $k$ values on the calculation results of the diffusion process, the following simulation experiment was conducted: assuming $D = 1$ m$^2$/s, the size of the simulation area is a square area of 11 m $\times$ 11 m, with the grid side length of 1 m. Initially, one unit of the relative concentration of pollutants was added to the center grid. Evidently, the equilibrium concentration is $1/121 = 0.0083$ units. With the simulation time steps of 0.2, 0.1, and 0.01 s, the corresponding $k$ values were calculated as 0.2, 0.1, and 0.01, respectively. The changes in the pollutant concentration in the center grid under various $k$ values are displayed in Figure 2.

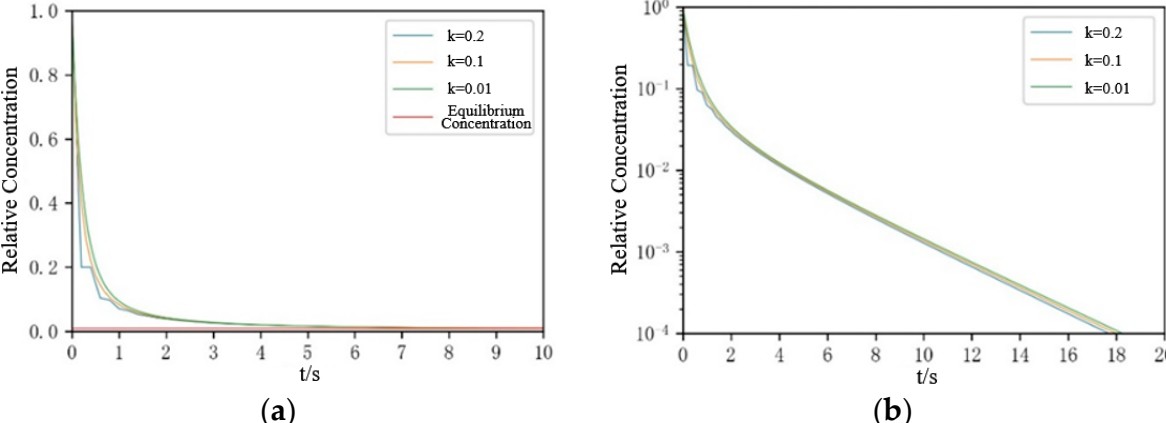

**Figure 2.** Variation curve of the pollutant concentration with time in the center grid with different *k* values. (**a**) Center grid concentration; (**b**) Difference between the center grid concentration and equilibrium concentration.

As shown in Figure 2a, all values reached the vicinity of the equilibrium concentration almost simultaneously. The two curves of *k* values of 0.1 and 0.01 almost completely overlapped, which indicated that a good result can be obtained when the *k* value is 0.1. The curve with a *k* value of 0.2 reveals a stair-like change in the early stage, which can be attributed to the interval between adjacent points (i.e., the simulation time step) being 0.2 s; however, the value still maintains an excellent degree of agreement with the other two curves in the subsequent stage.

The effects of molecular diffusion, turbulent diffusion, and dispersion should be considered while analyzing the diffusion coefficient $D$. Effects of both turbulent diffusion and dispersion are related to the local flow field and typically exhibit anisotropy. According to existing studies, the order of magnitude of the molecular diffusion coefficient in water is approximately $10^{-9} \mathrm{m}^2/\mathrm{s}$, the order of magnitude of the turbulent diffusion coefficient is approximately $10^{-2} \mathrm{m}^2/\mathrm{s}$, and the dispersion coefficient typically ranges between $10^2$ and $10^3 \mathrm{m}^2/\mathrm{s}$. Therefore, the effect of dispersion is typically predominant. The range of $D$ is $[10^2 \mathrm{m}^2/\mathrm{s}, 10^3 \mathrm{m}^2/\mathrm{s}]$, and the value of the horizontal diffusion coefficient $D$ is closely related to the grid side length.

### 2.2. Pollutant Transport Process

In the transport process, the directional movement of pollutants with ocean currents is considered, rather than the concentration changes. The assumption is that at the same time and location, the transport direction and velocity of the pollutants are consistent with those of the ocean current field. For a two-dimensional transport problem, the velocity of the flow field at the center of the grid is assumed to be $v$, with components along two orthogonal directions being $v_x$ and $v_y$. After the time step $\Delta t$, the fluid element at the center of the grid moves a distance of $v_x \Delta t$ and $v_y \Delta t$ in the x- and y-directions, respectively, thus reaching a new position. The concentration at the new position after transport is assumed to be equal to the concentration at the center of the grid before transport.

Because this method records only the pollutant concentration at the center of each grid, to maintain the grid division of the sea area, the moving distance should be an integer multiple of the grid side length $\Delta x$. When $\langle a \rangle = \lfloor a + 0.5 \rfloor$ (rounded to the nearest integer), the number of grids that the fluid element at the center of the small grid advances in the x- and y-directions can be denoted as $\langle v_x \Delta t / \Delta x \rangle$ and $\langle v_y \Delta t / \Delta x \rangle$, respectively. However, rounding of the data may cause distortion. For example, $\langle 1.6 \rangle$ and $\langle 2.4 \rangle$ are both equal to 2, though they differ considerably. To reduce the error caused by rounding and retain as much information about velocity magnitude and direction as possible, values of $\langle v_x \Delta t / \Delta x \rangle$ and $\langle v_y \Delta t / \Delta x \rangle$ should be large, or $v \Delta t / \Delta x$ should be >1.

Within a single time step $\Delta t$, pollutants diffuse to the adjacent grids, and the transport process causes the pollutants to move a distance of approximately $\langle v\Delta t/\Delta x\rangle$ grids. However, if $\langle v\Delta t/\Delta x\rangle$ is greater than 1, pollutants may not be able to diffuse in the direction against the current, resulting in a "vacuum area" in the pollution simulation, as displayed in the red area in Figure 3.

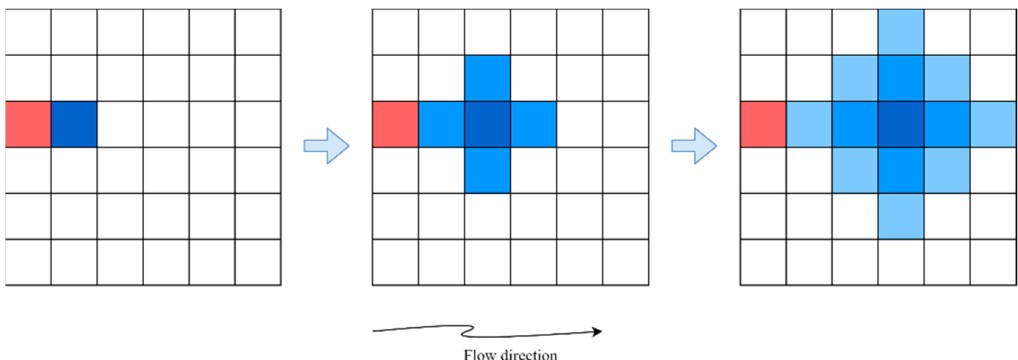

Flow direction

**Figure 3.** Schematic of "vacuum zone" in the simulated area. (The depth of the blue area indicates the level of concentration.)

This phenomenon contradicts the previous analysis requirement that $v\Delta t/\Delta x \gg 1$. To address this problem, this study used various time steps for the diffusion and transport processes. The time steps for the diffusion and transport processes are $\Delta t$ and $\beta\Delta t$, respectively, where $\beta$ is a positive integer. Here, $\beta$ only needs to satisfy the following conditions:

$$\begin{cases} \frac{\beta v\Delta t}{\Delta x} < \beta \\ \frac{\beta v\Delta t}{\Delta x} \gg 1 \end{cases} \text{ i.e., } \frac{1}{\beta} \ll \frac{v\Delta t}{\Delta x} < 1 \Rightarrow \Delta t < \frac{\Delta x}{v} \tag{7}$$

In this study, the maximum current velocity $v_{\max}$ was used instead of $v$ in the aforementioned equation for calculation.

By contrast, because of the deformability of the fluid, after the transport process, the concentration of multiple grid centers moves to the interior of the same grid. As displayed in Figure 4a, after transport, the concentration centers of grids $a$ and $b$ moved to grid $d$, and none of the concentration centers moved to grid $c$, leading to a situation where grid $d$ has multiple concentration inputs, but grid $b$ does not have the concentration input when calculating the concentration change during the transport process. Therefore, in this study, the inverse derivation method was used to calculate the concentration change in each grid center point during the transport process, i.e., for the center of each grid to be calculated, the fluid micro-element is calculated before time step $\beta\Delta t$. The pollutant concentration at this point is the concentration at the center of the grid after the pollutants are transported with ocean currents at each step. As displayed in Figure 4b, uniquely determined points ① and ② exist that correspond to the concentration center values of grids $c$ and $d$ to be obtained. This method ensures that each grid has unique concentration input data, in addition to avoiding the problem of missing data and better satisfying the analysis and calculation requirements of the transport process.

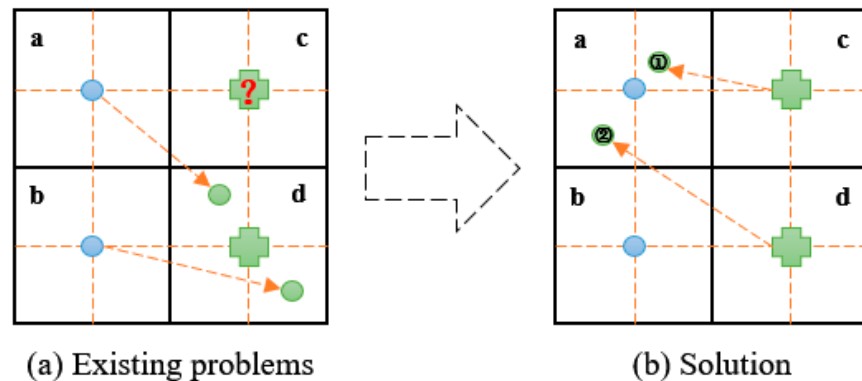

**Figure 4.** Grid concentration loss problem and solution diagram. (**a**) Existing problems; (**b**) Solution. (Arrows indicate migration paths of concentrations in the center of the grid. The blue and green parts represent the concentration before and after transport respectively.)

To achieve the balance between accuracy and efficiency, considering the previous exploration of $k$ value and the relevant constraints on the value of $D$, we determined the process of selecting the values of the diffusion coefficient $D$, simulation speed $k$, and diffusion process time step $\Delta t_k$. Figure 5 illustrates the value selection process for the diffusion coefficient $D$.

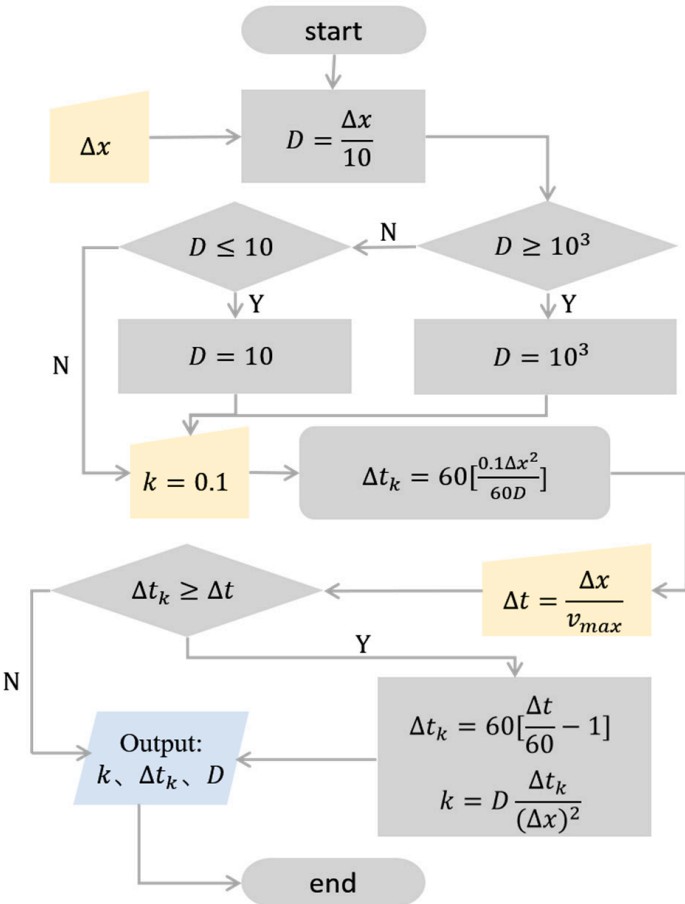

**Figure 5.** Process of selecting key parameters.

The process of selecting the values of various parameters can be simplified as follows: first, the horizontal grid side length $\Delta x$ is used to obtain the size of $D$ and determine whether $D$ is within the $[10\text{m}^2/\text{s}, 10^3\text{m}^2/\text{s}]$ interval. If the value exceeds the interval range, the interval endpoint value must be set as $D$. Furthermore, assuming $k = 0.1$ the value of

the diffusion process time step $\Delta t_k$ is calculated using Formula (6). Here, [ ] represents the rounding operation on the data for making $\Delta t_k$ a multiple of minutes for the convenience of subsequent simulation time calculations. Next, using Formula (7), the maximum time step value $\Delta t$ is calculated and then compared with $\Delta t_k$; if $\Delta t_k \geq \Delta t$, $\Delta t_k$ is recalculated using $\Delta t$, and subsequently, the $k$ value is recalculated, finally outputting the parameter values of the diffusion coefficient $D$, simulation speed $k$, and the diffusion process time step $\Delta t_k$.

When setting the horizontal grid side length $\Delta x$, completing the unit conversion from longitude to length is necessary. In the area of meridians with the same longitude, the distance of $1°$ of latitude was approximately 111 km; however, on the same latitude line, the actual distance of $1°$ of longitude was related to the latitude of latitude line $\alpha$, which is $\cos \alpha \times 111$ km. Therefore, in the subsequent analysis and simulation, the length of $1°$ of longitude is considered approximately $\cos \alpha \times 111$ km ($\alpha$ is the median latitude of the simulation area).

### 2.3. Pollutant Decay Process

For nuclear pollution problems in the ocean, the decay process typically considers the decay of radioactive elements. The decay process is related only to the elements and is independent of environmental conditions such as temperature and humidity. The process can be specifically expressed as follows:

$$\frac{dc}{dt} = -\lambda_{radio}c_0 \Rightarrow c = e^{-\lambda_{radio}t}c_0 \tag{8}$$

where $c$ and $c_0$ represent the concentrations of radioactive materials at time $t$ and the initial concentration, respectively; $\lambda_{radio}$ is the decay constant; and $t$ is the decay time (s). After time $\Delta t$, the concentration of radioactive materials becomes the original concentration multiplied by $e^{-\lambda_{radio}\Delta t}$. The relationship between the decay constant and the half-life $T_{1/2}$(s) is as follows:

$$\lambda_{radio} = \frac{\ln 2}{T_{1/2}} \tag{9}$$

For a single pollutant release, because the decay process calculation involves multiplying the concentration of all grid cells by $e^{-\lambda_{radio}\Delta t}$, and the diffusion process calculation is related only to the relative concentration values (though the transport process calculation is related only to the ocean current data), considering the decay process at every time step is not necessary. Instead, the difference in time $t$ between the start and end of the entire diffusion process should only be considered, and the calculation results of the concentration values at the center of each grid cell should be multiplied by $e^{-\lambda_{radio}t}$.

### 2.4. Data Architecture

In the case analysis stage, the multi-process fusion simulation analysis model involves input data, simulation process data, result analysis data, and visual data. Based on their attributes, the data can be classified into two categories, namely static and dynamic.

In terms of data sources, the model application data originates from three open-source datasets, namely the ETOPO1 dataset [38] jointly released by the U.S. National Geophysical Data Center (NGDC) and the U.S. National Oceanic and Atmospheric Administration, the Visible Earth dataset [39] funded by the National Aeronautics and Space Administration, and the HYCOM dataset [40] released by the HYbrid Coordinate Ocean Model Consortium (HYCOM). The specific data structure is displayed in Figure 6.

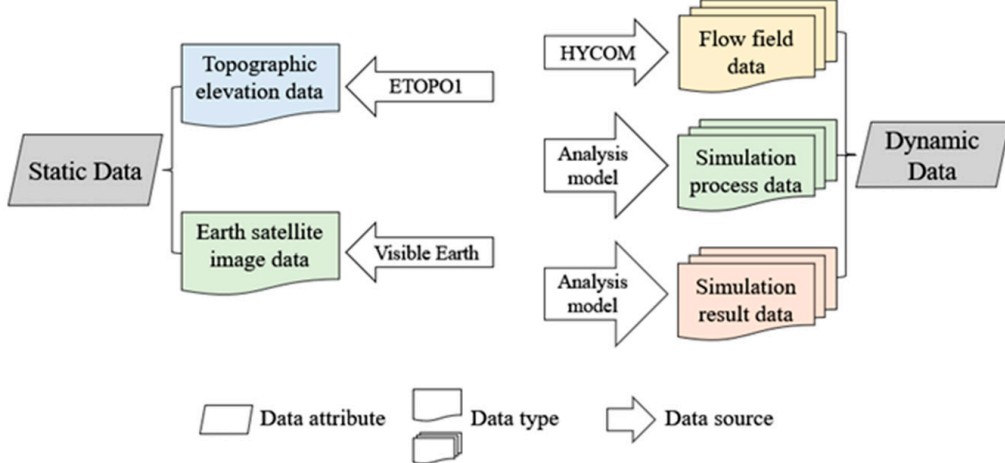

**Figure 6.** Process of selecting key parameters.

The data types and sources used in this study as follows:

1.  Topographic elevation data: These data were used to distinguish sea surface and land surface boundaries during the simulation process. The ETOPO1 dataset used in this study is the most accurate elevation data released by NGDC, with a resolution of 1 min;

2.  Earth satellite image data: Earth satellite images and background maps provide information for the simulation of the diffusion process. The data volume is small, and the dimension is not related to time. The required images can be directly extracted from the website of the Visual Earth Digital Library;

3.  Flow field data: This data type provides important support for diffusion simulation in local areas and serves as the ocean current velocity data source for the simulation analysis of the pollutant transport process. The HYCOM data set used in this study contains ocean surface current velocities from 80° S to 80° N, with the data accuracy of 0.04° and 0.08° in the latitudinal and longitudinal directions, respectively. The time interval was 3 h, and the data accuracy was higher than that of the other ocean current data. The reliability analysis shown in Figure S1 in the supplementary material showed that the accuracy of HYCOM ocean current data is higher in high-velocity areas near China. Therefore, for the case study, the areas with higher current velocities in the South China Sea (105° E to 116° E in longitude and 11.5° N to 23° N in latitude) were selected for simulation and analysis to ensure the accuracy of the simulation results.

### 2.5. Data Processing and Fusion Analysis

For the fusion analysis of the pollutant dispersion simulation, the C# programming language was used in this study to design the analysis process, as shown in Figure S2 in the supplementary material, which is conducive to the integration and expansion of the neutron process in the above method.

Accordingly, the sub-processes involved in Sections 2.1–2.3 were fused using the visual programming method to obtain the required simulation cloud diagram of the diffusion process. The specific process is displayed in Figure 7. The dataset information involved in Figure 7 is presented in Section 2.4.

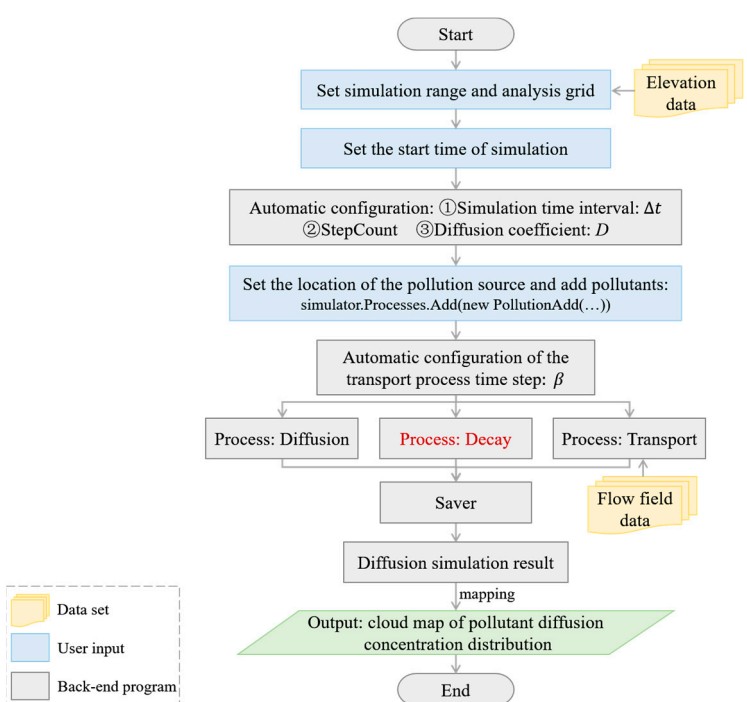

**Figure 7.** Implementation flow of pollutant diffusion simulation.

To satisfy the calculation requirements of diffusion simulation, first, encryption processing is performed to analyze the grid settings, and the bilinear interpolation algorithm is used as the encryption processing method for the flow field data grid. The bilinear interpolation algorithm is a commonly used algorithm for refining grid data, and it is used to estimate the points inside the grid using the function in the following formula:

$$v(x,y) = axy + bx + cy + d \tag{10}$$

where coefficients *a*, *b*, *c*, and *d* are obtained by substituting the data of the four corner points of the grid, with each grid generally with different coefficients. Using this method, for any given position, the ocean current speed can be calculated to satisfy the calculation requirements for the pollutant transport process.

Figure 7 shows the three sub-processes involved in the diffusion process. The subsequent diffusion simulation of organic pollutants can be extended by adding the corresponding process.

## 3. Case Study

### 3.1. Basic Situation

For the diffusion simulation analysis of heavy metal pollutants such as mercury and cadmium, as well as radioactive pollutants such as iodine-131 (half-life of 8 days) and strontium-89 (half-life of 50.5 days), the following four initial pollution source locations were selected in the study: (1) Daya Bay: 114.51° E, 22.42° N; (2) Hainan Houhai: 110.45° E, 18.72° N; (3) Dongshui Port: 110.08° E, 20.04° N; and (4) Dongchang Port: 109.86° E, 20.32° N (as displayed in Figure 8a). In the actual simulation process, because the diffusion analysis considers increments, the calculation process is related only to the relative values of pollutant concentrations. Assuming that no other pollution sources enters or leaves during the simulation process, the pollutant concentration of the initial pollution area increases by one unit at each time interval, and the pollutant concentrations in other areas at various times are calculated accordingly. Because the average depth of the South China Sea is only approximately 1212 m, which is considerably smaller than its length and width,

and the coastal waters are even shallower, this simulation process was approximated as a two-dimensional diffusion problem.

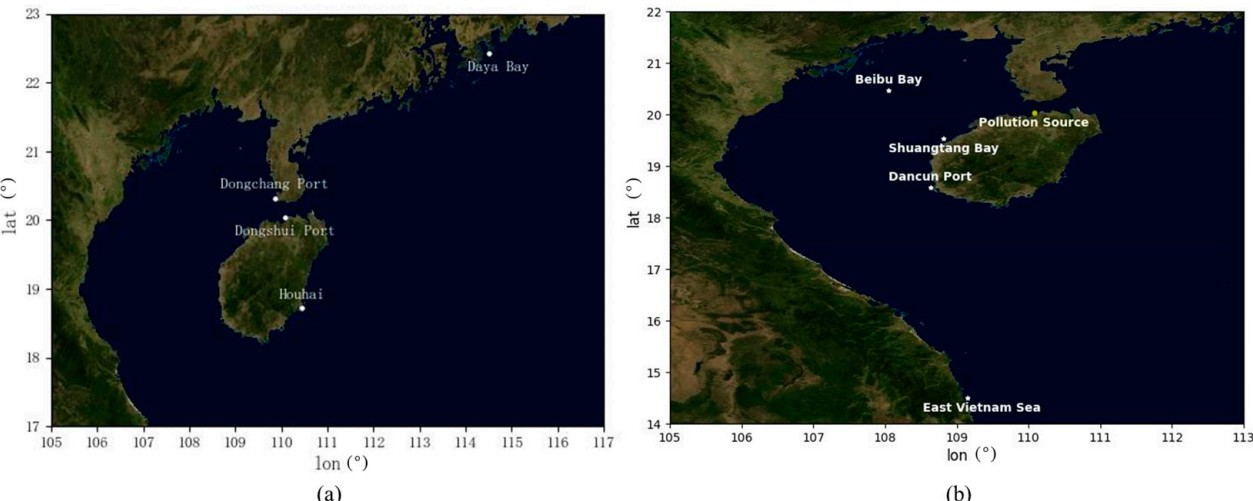

**Figure 8.** Schematic of the location of pollution sources and monitoring points. (**a**) Pollution source position; (**b**) Monitoring point position.

According to aforementioned requirements, the wastewater discharge is assumed to be continuous and periodic, i.e., pollutants are added to the initial pollution area once a day.

For the ocean current data used in the transport process, the aforementioned HYCOM data were used. South China Sea surface ocean current data for January 2023 were extracted from the HYCOM dataset, and a simulation range of 105° E to 116° E in longitude and 11.5° N to 23° N in latitude was used to calculate concentration change during the transport process. According to a brief analysis of the data, the maximum surface ocean current velocity in the simulation area is $|v_{\max}| = 1.46 \, \text{m/s}$ . Figure 9 displays the ocean current data in the simulation area at various time points. The ocean current velocity is distinguished by different colors in the legend, and the gray area represents the land boundary (the gray dots in the sea surface area may be island boundary or data missing area).

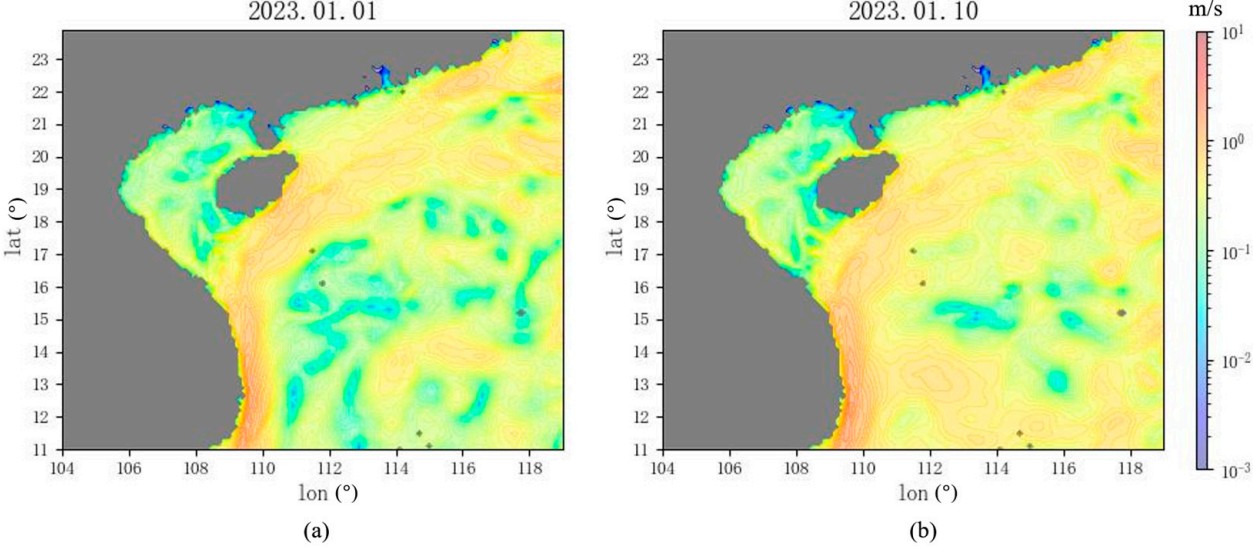

**Figure 9.** Sea surface current velocity data at various times, obtained from the HYCOM dataset. (**a**) Time point: 2023.01.01; (**b**) Time point: 2023.01.10.

To divide the analysis area, the grid side length of the simulation area was assumed to be $0.01°$, which is $\Delta x = 0.01 \times \cos\alpha \times 111 \times 10^3 = \cos((23.5 + 13.5)/2) \times 111 \times 10 \approx 1060$ m, with the initial pollution area occupying one grid. Based on the parameter value selection process in Figure 5, the simulation time step should satisfy the following requirements:

$$\frac{v_{max}\Delta t}{\Delta x} < 1 \Rightarrow \Delta t < \frac{1060 \text{ m}}{1.46 \text{ m/s}} = 726.1 \text{ s} \tag{11}$$

Considering $D = \Delta x/10 \approx 106, k = 0.1 \Rightarrow \Delta t_k = 60\left[k\Delta x^2/(60D)\right] = 1080 \text{ s} > \Delta t$; thus, taking $\Delta t_k = 60[\Delta t/60 - 1] = 660$s $\Rightarrow k = D\Delta t_k/\Delta x^2 = 106 \times 660 \div 1060^2 = 0.062 < 0.2$, which satisfy the requirement of calculation convergence in the diffusion process. The event step coefficient $\beta$ in the transport process should satisfy the following requirements:

$$\frac{1}{\beta} \ll \frac{v_{max}\Delta t_k}{\Delta x} \Rightarrow \beta \gg \frac{\Delta x}{v_{max}\Delta t_k} = \frac{1060}{1.46 \times 660} = 1.10 \tag{12}$$

Considering $\beta = 90$, the time step of the transport process is 16.5 h. When simulating the transport process, the displacement should not be calculated in one step according to $v\beta\Delta t$, but rather the sum of $\beta$ times $v\Delta t$, where $v$ changes with the position (cumulative displacement). This phenomenon can avoid the distortion of results due to the large time step in the transport process.

### 3.2. Simulation Results Analysis

According to the proposed pollutant diffusion model and visualization technology, the decay sub-process components of pollutants are closed. When not considering other biological factors and vertical transport in the ocean, Figure 10 depicts the simulation results of pollutant diffusion of heavy metals such as mercury and cadmium (which does not degrade easily in the ocean) obtained through programming (with $10^{-12}$ as the minimum concentration unit). The locations of the pollution sources are displayed in Figure 8a, and the names of the monitoring points involved are detailed in Figure 8b.

According to the simulation results in Figure 10, the pollutants diffuse rapidly in the early stage. After the wastewater is released in the initial pollution area, it is affected by ocean currents, and the diffusion speed of pollutants along the meridian direction is considerably greater than that along the parallel direction, eventually accumulating along the coast. Comparing the simulation results in the emission scheme (a) shown in the red box with each emission sub-process reveals that the expansion of pollutant area is affected by the diffusion sub-process, and the transport sub-process plays a crucial role in the long-distance migration of pollutants. The diffusion rate of pollutants is considerably affected by ocean current transport, whereas the change in concentration is affected mainly by diffusion. The final simulation result map was obtained by the superposition of the position changes in each sub-process, rather than the superposition of the concentration change, indicating that the final pollutant diffusion simulation result map cannot be obtained simply by superimposing the simulation maps of each sub-process.

Comparing emission schemes (b) and (c) revealed that although the concentration center of the pollutants was near the emission point in the initial stage, with time, the high-concentration area of the pollutants (yellow and red parts) extends southwestward along the 109° E meridian, spreading from the initial area near the South China Sea to the area near the eastern Vietnamese Sea. Comparing emission schemes (c) and (d) revealed that even if the pollution outlet locations are set close to each other and only separated by the sea, the range of the core area of pollutant diffusion concentration differs considerably because the current speed in Beibu Bay is slow. When the pollution outlet was set near Dongchang Port, the pollutants were affected by its current speed and closed-loop ocean currents, resulting in pollutants accumulating in the bay and only a small portion of the pollutants flowing into the open sea being diluted by ocean currents. However, when the pollution outlet was set near Dongchang Port, the surrounding current speed was fast,

rendering the pollutants easier to dilute, and only a small part spread northward into Beibu Bay.

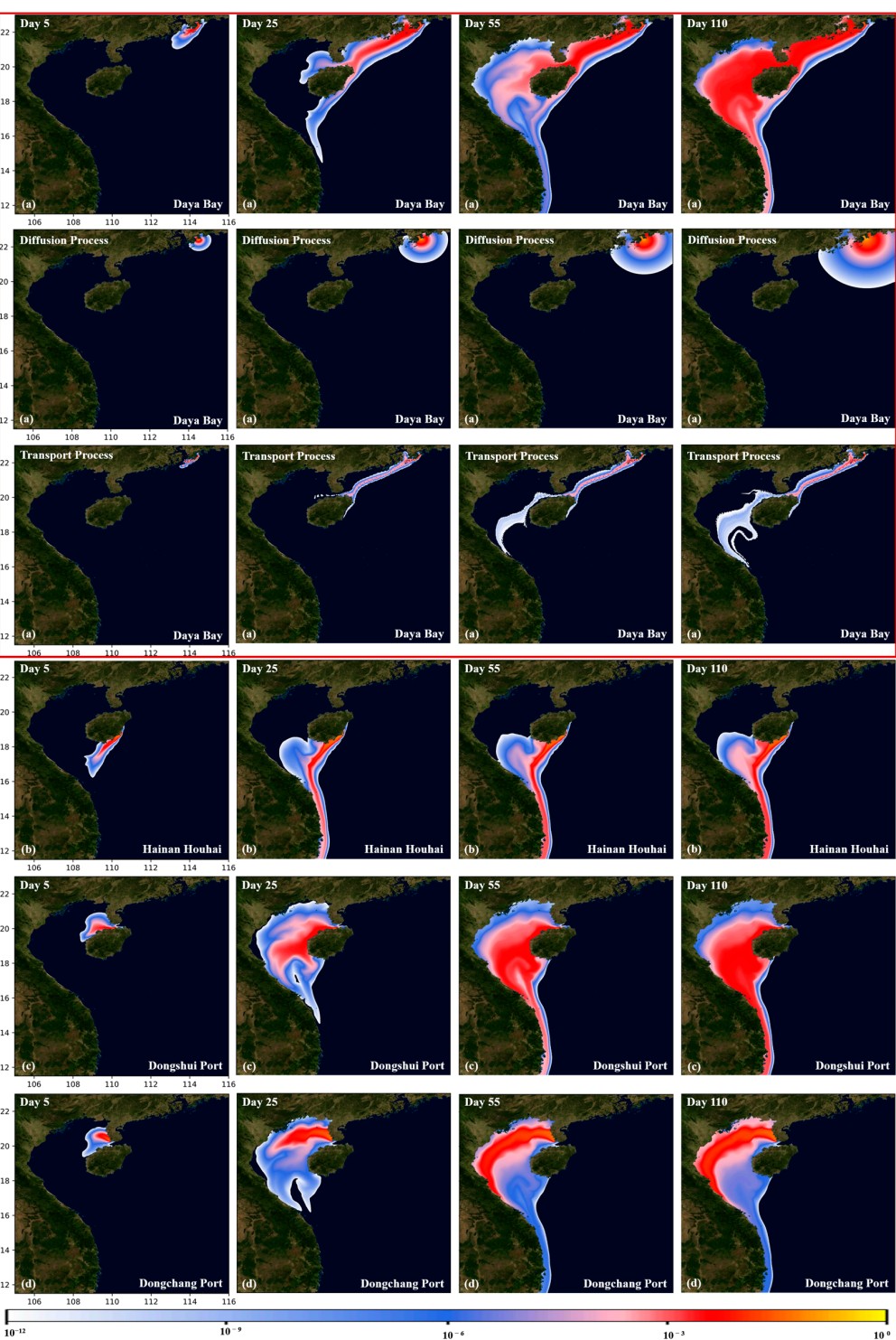

**Figure 10.** Simulation results of heavy metal pollutant diffusion. (**a**) Pollution source: Daya Bay (Top: Simulation result; Middle: Diffusion process; Bottom: Transport process); (**b**) Pollution source: Hainan Houhai; (**c**) Pollution source: Dongshui Port; (**d**) Pollution source: Dongchang Port.

To compare the concentration changes in various pollution sources at a particular monitoring point, the concentration changes in pollution sources near Dongshui Port and Dongchang Port at the Beibu Bay monitoring point were compared; the results are

displayed in Figure 11a. In addition, the concentration changes in pollution sources near Dongshui Port and Hainan Houhai at the eastern Vietnamese Sea monitoring point were compared, as illustrated in Figure 11b. The names of the pollution sources and monitoring points are displayed in Figure 8a,b, respectively.

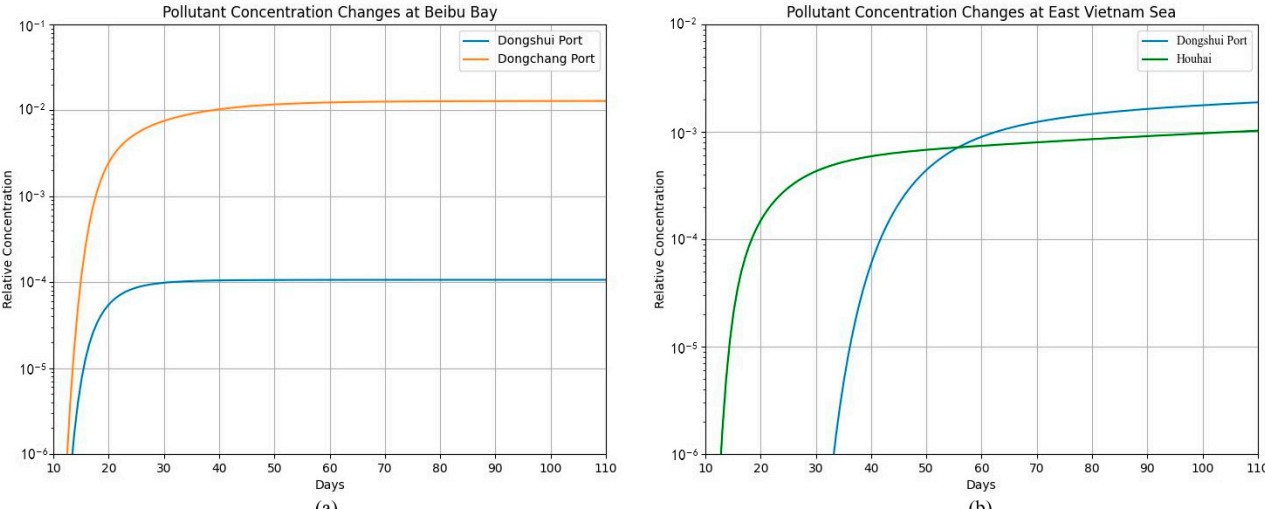

**Figure 11.** Comparison of the pollutant concentration of various pollution sources at the same measuring point. (**a**) Pollutant concentration changes at Beibu Bay; (**b**) Pollutant concentration changes at East Vietnam Sea.

Figure 11a reveals that the accumulation of pollutants from the Dongchang Port pollution source is more significant because of the slower ocean current exchange speed in Beibu Bay. In the early stage of diffusion, the pollutant concentration from the Dongchang Port pollution source increased considerably, and although the time required to reach equilibrium concentration is approximately 20 days later than that of the Dongshui Port pollution source, the equilibrium concentration is approximately 100 times higher.

Figure 11b shows that although the Houhai pollution source is closer to the eastern Vietnamese Sea monitoring point, the time required for the concentration to reach $10^{-7}$ units at the monitoring point was nearly 30 days earlier than that for the Dongshui Port pollution source. However, the increase rate of the Dongshui Port pollution source at the monitoring point was considerably higher than that of the Houhai pollution source. If the same number of pollutants was discharged from the two pollution sources, the equilibrium concentration of the Dongshui Port pollution source at the monitoring point would be greater than that of the Houhai pollution source.

If the daily discharge and heavy metal pollutant content of each discharge outlet is known, the heavy metal pollutant content in the nearby sea area within a specified time range can be calculated, which can provide corresponding technical support for pollution prevention and control.

To provide a clear comparison and analysis of the pollution situation in various areas within the simulation range, in this study, representative monitoring points within the simulation range were selected, and the changes in the relative concentration of pollutants at each monitoring point over time for the emission scenario (c) were extracted. The locations of pollution sources and monitoring points are displayed in Figure 8, and the comparison results are presented in Figure 12a.

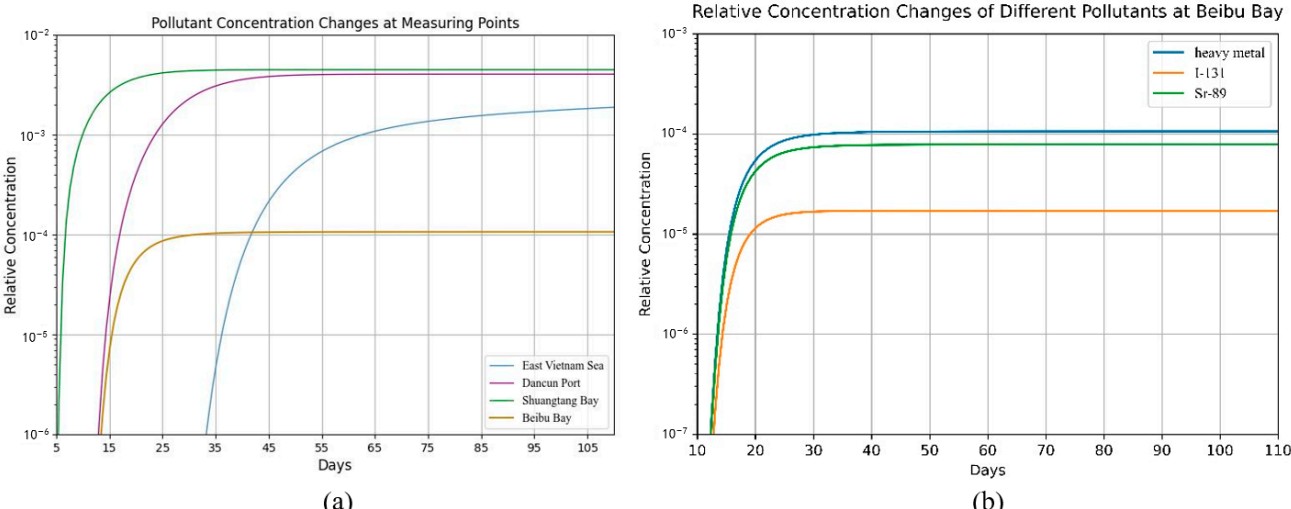

**Figure 12.** Relative concentration changes: (**a**) Pollutant concentration changes at monitoring points; (**b**) Relative concentration changes in various pollutants at Beibu Bay.

The concentration change curves of different monitoring points (Figure 12a) reveal some common features. When the concentration of pollutants is low, the growth rate is very fast, whereas, with an increase in the concentration, the growth rate gradually slows down. Some notable differences were observed in the concentration change curves of each monitoring point. In terms of the order of appearance of pollutants, although the Shuangtang Bay monitoring point is closer to the pollution source, its early pollutant concentration change and stable concentration value were similar to those of the Dancun Port monitoring point. Furthermore, although the time for the concentration value of the eastern Vietnamese Sea monitoring point to reach $10^{-6}$ units is approximately 20 days later than that for the other monitoring points, its growth rate is considerably higher than that of the Beibu Bay monitoring point, overtaking it around day 41. When the concentration at this monitoring point reached equilibrium, it is possibly higher than the equilibrium concentration of the Beibu Bay monitoring point.

The comparative analysis revealed that even if the pollutant concentration increased rapidly in the early stage of diffusion, it did not necessarily indicate that the final equilibrium concentration was higher. This approach can help us develop appropriate monitoring and prevention measures based on the equilibrium concentration and equilibrium time of pollutants at a specific monitoring point. Considering the pollution source near Dongshui Port as an example, while analyzing the worst impact on the nearby sea area of the Dancun Port monitoring point, the pollutant concentration values around 55 days after emission should be estimated. However, while analyzing the worst impact on the nearby sea area of the eastern Vietnamese Sea monitoring point, we should estimate the values when the pollutant concentration reaches a dynamic equilibrium near the monitoring point after 110 days, without necessarily exploring the overall diffusion process.

Furthermore, using the relevant parameter settings in Section 3.1, except for the types of pollutants, the rest of the simulation conditions were controlled, the pollutant decay sub-process component was opened, and the diffusion of heavy metal pollutants that are difficult to degrade, such as radioactive pollutants (iodine-131, and strontium-89), was compared. The comparative analysis results are displayed in Figure 12b, and the concentration distribution is depicted in Figure 13.

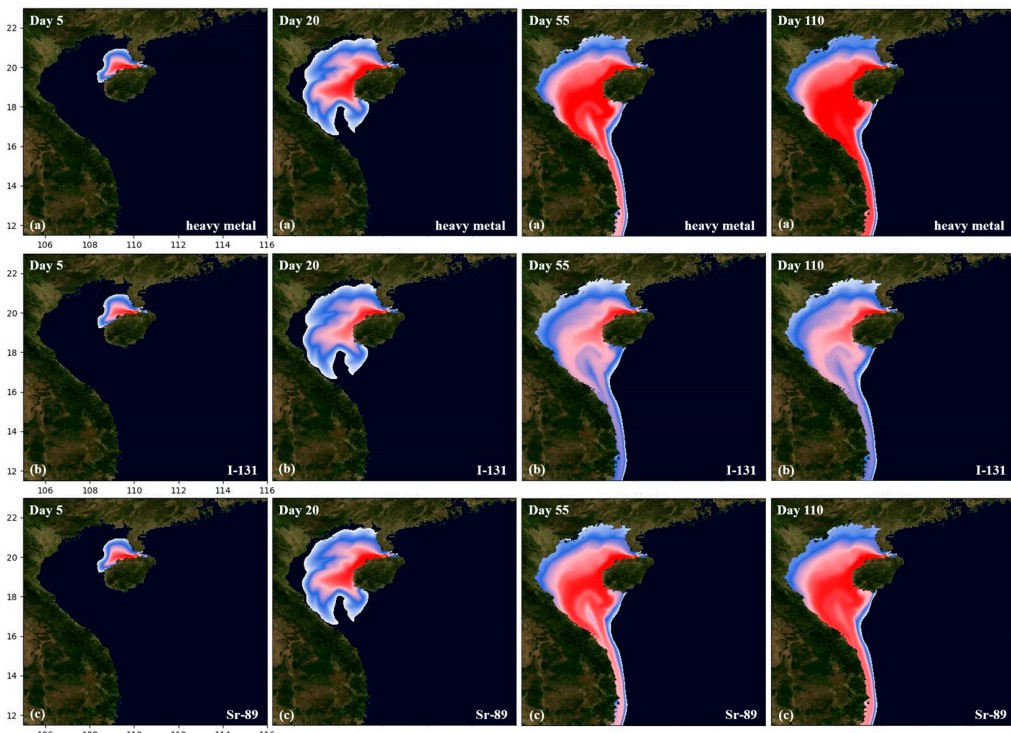

**Figure 13.** Concentration distribution of different pollutants. (**a**) heavy metal; (**b**) I-131; (**c**) Sr-89.

According to the comparison analysis charts and simulation results of pollutant diffusion, the diffusion paths, diffusion speeds, concentration centers, and final diffusion ranges of different pollutants released from the same pollution source were consistent. However, the equilibrium concentrations of the pollutants differed. Figure 12b reveals that the equilibrium concentrations of the different types of pollutants are typically affected by their half-life. To conclude, under the same simulation conditions, the shorter the half-life of marine radioactive pollutants, the lower their equilibrium concentrations in the ocean water. However, the half-life and equilibrium concentration of the pollutants do not exhibit a linear relationship.

## 4. Discussion and Limitations

### 4.1. Discussion

This study proposes a multi-process fusion pollutant diffusion numerical simulation method for typical marine pollutants based on flow field data. This method offers the following advantages:

1. The diffusion process of pollutants was categorized into several independent sub-processes, thus expanding the scope of the analysis of the method is easy. For example, although this study considered the diffusion of a single pollution source in the South China Sea, the proposed method can be used for the diffusion analysis of multiple pollution sources. Because the diffusion processes are independent of each other, the simple superposition of each pollution source's diffusion process is sufficient. Simultaneously, the proposed simulation method can also involve more sub-processes. For example, by introducing a microbial degradation coefficient after the completion of the diffusion and transport process simulations, the diffusion process of N, P, and other organic matter can be analyzed, thus providing good scalability.

2. The proposed pollutant diffusion simulation method can be used for simulating any global marine area based on HYCOM data. Moreover, the method does not depend on specific ocean current models, and the ocean current dataset with arbitrary precision can be replaced according to the needs of the study, rendering the simulation process simple and efficient.

3.  In this study, the calculation process was simplified using grid division, diffusion velocity $k$ value, and diffusion coefficient $D$, therefore eliminating the need for complex calculations, such as differentiation during the solving process. Hence, this method can reduce the number of calculations without considerably decreasing accuracy, therefore saving computational resources and shortening the simulation time.

### 4.2. Limitations

Because of the limitations of time and energy, the scope of further improving the proposed marine pollutant diffusion model exists. The limitations of the model are reflected in the following aspects:

1.  The pollutant diffusion model proposed in this study is based on the assumption of planar diffusion and does not require the three-dimensional diffusion method, because the sea area near China is shallow, the inshore water depth is shallower, and the ocean depth is negligible compared with the plane simulation range. However, the disadvantage of this method is that the vertical transport of pollutants and the interaction between seafloor sediments are not considered, which decreases the accuracy of the model.

2.  In the key settings, this study considered some influencing factors of the diffusion coefficient $D$; however, in practice, this coefficient is affected by ocean currents, temperature, and other factors, and their value should be determined through further experiments.

3.  This study assumed that the sub-processes of the pollutant diffusion model are independent of each other; however, because the ocean is a typical complex system, the sub-processes may interact with each other and influence the actual pollutant diffusion processes, such as suspension adsorption, and the interaction between the atmosphere and the upper surface layer of the ocean. Furthermore, internal sinks, i.e., the enrichment and migration of marine organisms exert a certain degree of influence on the diffusion of pollutants. Specifically, for the diffusion of radioactive pollutants in the ocean, the effect of biological factors may be greater than that of the transport of ocean currents. Therefore, the negative effects on human beings cannot be determined only based on the pollutant concentration in seawater. Subsequently, relevant influence factors should be added or the corresponding biological form attenuation term should be set in the model.

### 4.3. Future Directions

In the follow-up study, we aim to conduct in-depth research on the limitations mentioned in Section 4.2 and optimize the pollutant diffusion model from the following three aspects:

1.  Modifying the two-dimensional diffusion analysis method to a three-dimensional diffusion method: In subsequent studies, the three-dimensional flow field at the ocean scale can be constructed using ocean current data at various depths in the HYCOM flow field data, which can support the application of the proposed method in three-dimensional diffusion simulations.

2.  Optimizing the marine pollutant diffusion model: The parameter value process should be further optimized, and the actual emission process should be tested and monitored to optimize the key parameter values of the model. Moreover, for higher accuracy simulation requirements, using higher-order differential equation-solving methods or more refined grid division methods for optimizing the analysis methods of each sub-process should be used in further studies. Furthermore, more appropriate ocean current models or data generation from monitoring should be considered for specific areas of the ocean.

3.  Considering the influence of more factors in the ocean: Future studies can consider exploring the environmental characteristics of the ocean comprehensively and sub-dividing the overall diffusion sub-process of different pollutants. Additionally, the

influence of biological enrichment, coastal accumulation, vertical transport, air-sea interaction, and other factors on the simulation process should be considered.

## 5. Conclusions

By analyzing the diffusion process of typical pollutants in the ocean, a large-scale pollutant fusion simulation analysis model based on the concentration analysis method is proposed in this study, and key technologies such as the parameter value, data selection, visualization, etc., were investigated. Through the iterative analysis and programming implementation of three independent sub-processes, such as diffusion and transport of pollutants, this study visually presents the drift process of heavy metal and radioactive pollutants in common industrial and agricultural wastewater in the ocean. Moreover, the scientific analysis provides data and model support for the formulation of relevant emission policies.

Based on the methods, Section 3 of this paper presents the simulation results of the pollutant diffusion of four pollution sources near the South China Sea. The simulation results revealed that because of the influence of ocean currents, the diffusion speed of pollutants in the zonal direction was considerably greater than that in the meridional direction, and eventually, the pollutants tend to accumulate along the coast. The comparison of the simulation results in the emission scheme (a) shown in the red box with each emission sub-process revealed that the expansion of pollutant area is considerably affected by the diffusion sub-process, and the transport sub-process plays a leading role in the long-distance migration of pollutants. The diffusion rate of pollutants is considerably affected by ocean current transport, whereas the change in concentration is affected by diffusion. The final simulation result map was obtained by superpositioning the position changes in each sub-process, rather than the concentration change, implying that the final pollutant diffusion simulation result map cannot be obtained simply by superimposing the simulation maps of each sub-process.

Comparative analysis of the concentration changes in pollutants from the Dongshui Port pollution source at different monitoring points and pollution sources at the same monitoring point revealed that even if the pollutant concentration increased rapidly in the early stage of diffusion, the final equilibrium concentration may not be high. We can formulate corresponding monitoring plans based on the equilibrium concentration and equilibrium time of pollutants at a certain monitoring point.

Furthermore, according to the analysis of the diffusion of different pollutants, the decay process of pollutants only affects their equilibrium concentrations. Under the same simulation conditions, the shorter the half-life of radioactive pollutants, the lower their equilibrium concentrations in marine waters, with basically no effect on the diffusion path and final diffusion range.

Section 4 of the article discusses the advantages of the proposed method as well as its limitations and future research direction.

The proposed method in the present study provides reliable technical support for the dynamic tracking of marine pollutants.

**Supplementary Materials:** The following supporting information can be downloaded at: https://www.mdpi.com/article/10.3390/su151310547/s1, Figure S1: Implementation flow of pollutant diffusion program; Figure S2: Algorithm flow of Pollutant Diffusion Simulator.

**Author Contributions:** Conceptualization, X.G. and Y.L.; Methodology and visualization, X.G. and Y.L.; Investigation and writing—original draft, X.G.; Writing—review and editing, Z.H. and S.C.; Resources and supervision, J.-M.Z., S.C., S.L. and Z.-Z.H. All authors have read and agreed to the published version of the manuscript.

**Funding:** This research was funded by the Guangdong Basic and Applied Basic Research Foundation (2022B1515130006), the Major program of stable sponsorship for higher institutions (Shenzhen Science and Technology Commission, WDZC20200819174646001), and the Shenzhen Key Laboratory of Marine IntelliSense and Computation (ZDSYS2020081142605016).

**Institutional Review Board Statement:** Not applicable.

**Informed Consent Statement:** Not applicable.

**Data Availability Statement:** The data presented in this study are available on request from the corresponding author.

**Conflicts of Interest:** The authors declare no conflict of interest.

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
