# Peer review of "Simulation Analysis of the Dispersion of Typical Marine Pollutants by Fusion of Multiple Processes"

_sustainability, doi:10.3390/su151310547_

Round 1

Reviewer 1 Report

Interesting approach. I understand the notion that because diffusion is based on relative differences in concentration, and the radioactive decay is essentially the same everywhere, and hence, the inherent disconnectedness is not a serious issue. But this is also predicated on their being no other form of "decay" to this material (ie., internal sinks such as a flora or fauna that gets imbued with the radioactive material or heavy metal that leaves the grid cell). Please discuss the limitations of your approach in terms of how realistic it could or can't be. 

In general the paper could use with a more conventional structure. It needs a clearly stated objective and a clearly stated methodology. Section 2 is obviously a methodology as is section 3 up to 3.2 but 3.3 seems to start showing results which should be a separate section. I understand 3.3 is illustrating why that data set was used but Figure 7 should be in a supplementary data file as it is not imperative to the results which is the case study in section 4. The methodology section should note the study site used in the case study (that information is currently in section 4) as well as the data used, mathematical methods in section 2 and 3, etc.

Line 22 - 56 needs references to back up the statements (such as in the Minamata disease incident in Japan, etc).

Paragraphs 58-73 are in the wrong place. They sound like an objective followed by a brief methodology. They should be moved after the literature review which is finally seen lines 87 to 146. Please move those two paragraphs to be shown after line 146 and just before section 2. I would also suggest reducing the opening three paragraphs in lines 22-56 to be more succinct and reduced in length, particularly if you choose not to insert many references. I might suggest deleting the paragraph you currently have before section 2 and shown in lines 147 - 153 as it is somewhat repetitive.

Line 195 reads: Before and after the pollutants pass through the contact surface of the grid, the concentration changes of the two grids are both:...This statement is badly formed. Please clarify exactly what equation 4 is supposed to say - do you mean after pollutant pass from gridcell/square 1 to gridcell/square 2? If you mean that the increase in pollutant mass in cell 2 is equivalent to the decrease in pollutant mass in cell 1 than say this or at least reword the sentence to be clearer and let's cumbersome.

Line 203, what does this mean?: "Since there are usually 4 grids around each grid, to ensure the convergence of results," If you mean spatially surrounding each grid cell, there are eight grid cells. Please clarify that you mean that no diagonal movement is possible in your representation.

Line 204, why is this so? :"it must satisfy k <= 1/ (4 +1) = 0.2" Can you explain this a little more clearly.

Lines 318 to 338 begin to speak of software specific terms like Handel, Process object, StepCount. Unless there is specific information in the supplementary info (if provided by the authors) that detail what they are, I recommend removing this section. The paper is long and specifics related to software are not necessary. Ask yourself, what does this paragraph convey and is it necessary to the science? Remove it to supplementary information if you don't want to eliminate it.

How is the elevation data used shown at the top of Figure 6? What are these data? Section 346 to 357 is a good idea to keep if the authors delete the preceding material as requested above.

In lines 372 to 380, current data sets are compared with the suggestion that Hycom is more accurate. Unless you state what measure you used to compare the two (RMS for example) and what those values were, please place this entire section in a supplementary data.

Lines 383 to 386 should be in the methodology.

In figure 9 both illustrations show dots. What are these dots? This figure should be labeled with (a) and (b) and each should be fully described. The caption is currently too brief. The legend has no units.

Your simulation in 4.2 of heavy metals declares is non-degradable. Heavy metals may not "degrade" as in how the radioactive components are modeled but there are certainly sinks such as calcium carbonate materials in the nearshore area that would uptake these metals. No assumptions of this are stated nor is whether or not you turned off the decay component of the model.

Because this effectively a numerical exercise with no validation of the contaminant concentrations from in situ data sources, I would strongly recommend the Authors parse out the diffusion versus advection component of the modelling to better illustrate which process is dominant. Normally in a region like this, advection dominates and diffusion may be negligible. Can the Authors add another figure demonstrate the relative contributions to the concentrations of these two components in the heavy metal simulation (assuming there is no decay considered).

Line 434 starts with: "From the simulation results in Figure 10, it can be seen that the pollutants diffuse rapidly in the early stage. After the wastewater is released in the initial pollution area, it is affected by ocean currents, and the diffusion speed of the pollutants along the meridian direction is significantly greater than that along the parallel direction, eventually accumulating along the coast." Hence it would be good to show the relative contributions of the diffusion portion versus the current transport portion (the advection) in the simulation. In line 478 further discussion of diffusion is consistently mentioned but is there any real understanding of how much diffusion plays a role in this simulation? To figure 12 and 11b, I would partition each curve into a diffusion component and an advection component.

Figure 13 is one step toward demonstrating the role decay has in the three process simulation (and it's all dependent on the half life). The results are somewhat obvious but what's missing in all the simulations are the starting concentrations. Heavy metal concentrations surely would be much higher concentrations than radioactive materials - did the simulation start with the same concentrations for all simulations? Information on the simulation initial conditions needs to be added in the methodology section. While line 389 says this: "Assuming no other pollution sources enter or leave during the simulation process, the initial pollution area's pollutant concentration is increased by 1 unit each time, and the pollutant concentrations in other areas at different times are calculated accordingly." Does that mean the initial concentration is just 1 unit for all contaminants and increasing by 1 unit at each time step? Why didn't you start with a large single influx with a steady release? Wouldn't that have been more realistic?

The English writing is just fine.

Author Response

Thank you again for all the comments. Now we think the manuscript has been improved a lot and we have asked a professional language service to proof read and improve language of the manuscript, hoping to meet your expectations.(See the attachment for the details of the revision)

Reviewer 2 Report

The manuscript entitled Simulation Analysis Method of Typical Marine Pollutant Dispersion with Fusion of Multiple Processes, by X.-Q. Guo, Yi Liu, J.-M. Zhang, S. Chen, S. Li and Z.-Z. Hu, presents an interesting work.

In general, the manuscript should be acceptable for publication but some serious problems must be repaired prior to publication. It needs some significant improvement. Some suggestions are as follows:

  1. Please follow the journal author instructions. It would be useful for the reader to follow the classical text structure (i.e. Introduction-methodology-results-discussion-conclusions). A better presentation of your results and an extensive discussion would improve your paper.
  2. Please use different terms in the “Title” and the “Keywords”.
  3. The abstract should state briefly the purpose of the research, the principal results and major conclusions. An abstract is often presented separately from the article, so it must be able to stand alone.
  4. The English language usage should be checked by a fluent English speaker. It is suggested to the authors to take the assistance of someone with English as mother tongue.
  5. You could incorporate a map of study area.
  6. You could enrich the scientific literature.
  7. The authors could take into account the following publication: Makri, P.; Stathopoulou, E.; Hermides, D.; Kontakiotis, G.; Zarkogiannis, S.D.; Skilodimou, H.D.; Bathrellos, G.D.; Antonarakou, A.; Scoullos, M. The Environmental Impact of a Complex Hydrogeological System on Hydrocarbon-Pollutants’ Natural Attenuation: The Case of the Coastal Aquifers in Eleusis, West Attica, Greece. J. Mar. Sci. Eng. 2020, 8, 1018. https://doi.org/10.3390/jmse8121018
  8. Correct references in the text and the reference list according to the journal’s format. Please format the references’ list by using the correct journal abbreviations. See the following link: https://images.webofknowledge.com/images/help/WOS/A_abrvjt.html

Moderate editing of English language required

Author Response

(The authors gave the same response as above.)

Round 2

Reviewer 1 Report

Good job on the revisions.

There are still some minor typographical and grammatical errors, but those can be corrected by the copyright editor.

Reviewer 2 Report

This manuscript presents an improved work.

The manuscript should be acceptable for publication as it is.